# StoryAlign: Evaluating and Training Reward Models for Story Generation

**Haotian Xia,**[*] **Hao Peng**[*]**, Yunjia Qi, Xiaozhi Wang, Bin Xu,**[†] **Lei Hou, Juanzi Li**
Department of Computer Science and Technology, Tsinghua University
{xiaht24,peng-h24}@mails.tsinghua.edu.cn
{xubin, lijuanzi, houlei}@tsinghua.edu.cn

## Abstract

Story generation aims to automatically produce coherent, structured, and engaging narratives. Although large language models (LLMs) have significantly advanced text generation, stories generated by LLMs still diverge from human-authored works regarding complex narrative structure and human-aligned preferences. A key reason is the absence of effective modeling of human story preferences, which are inherently subjective and under-explored. In this work, we systematically evaluate the modeling of human story preferences and introduce STORYRMB, the first benchmark for assessing reward models on story preferences. STORYRMB contains $1,133$ high-quality, human-verified instances, each consisting of a prompt, one chosen story, and three rejected stories. We find existing reward models struggle to select human-preferred stories, with the best model achieving only $66.3\%$ accuracy. To address this limitation, we construct roughly $100,000$ high-quality story preference pairs across diverse domains and develop STORYREWARD, an advanced reward model for story preference trained on this dataset. STORYREWARD achieves state-of-the-art (SoTA) performance on STORYRMB, outperforming much larger models. We also adopt STORYREWARD in downstream test-time scaling applications for best-of-n (BoN) story selection and find that it generally chooses stories better aligned with human preferences. We will release our dataset, model, and code to facilitate future research. Related code and data are available at https://github.com/THU-KEG/StoryReward.

## 1 Introduction

The task of story generation aims to automatically generate coherent, structured, and engaging narratives from a given premise (Wang et al., 2023). With the rise of large language models (LLMs) and their powerful text generation capabilities, many studies leverage them and design workflows to generate fluent stories (Alhussain & Azmi, 2021; Wang et al., 2024b; Lee et al., 2025; Xia et al., 2025; Huot et al., 2025). However, LLM-generated stories are often considered to lack complex narrative structure and creativity (Chakrabarty et al., 2024; Tian et al., 2024; Wang & Kreminski, 2025). For example, Chakrabarty et al. (2024) report that LLM-generated stories are three to ten times less likely to pass expert evaluation compared with those written by professional authors. This gap highlights that, despite significant progress, LLM-generated narratives remain far from human standards and preferences.

The main reason is that current LLMs lack effective modeling of human story preferences during training. LLMs typically rely on reinforcement learning from human feedback (RLHF; Ouyang et al., 2022), which first trains a reward model (RM) to capture human preferences and then optimizes LLMs based on these rewards. However, existing reward models mainly focus on general domains (Zhong et al., 2025) and may overlook story-specific preferences. Meanwhile, they also exhibit biases, such as verbosity bias (Saito et al.; Peng et al., 2025), which favors longer rather than higher-quality responses. As a result, models trained with such reward signals may inherently struggle to generate stories aligned with human preferences. Moreover, modeling story preferences is inherently challenging,

---

[*] Equal contribution.
[†] Corresponding author.

as it is subjective and spans multiple dimensions such as creativity and relevance. The automated evaluation of LLM-generated stories thus remains an open problem (Yang & Jin, 2024). Therefore, effectively evaluating and improving story preference modeling of RMs is necessary.

To address this gap, we systematically evaluate the modeling of human story preferences and develop an advanced reward model for story generation. Specifically, we introduce STORYRMB, the first benchmark for evaluating reward models of story preferences. Inspired by prior work (Yang & Jin, 2024), we define 5 evaluation criteria for STORYRMB, including coherence, creativity, characterization, fluency, and relevance. The construction of STORYRMB consists of two parts: (1) Candidate story generation, which collects diverse stories for the same premise from 4 advanced LLMs, including Gemini 2.5 Pro (Comanici, 2025), Grok 3 (xAI, 2025), GPT-4o (OpenAI, 2024), Qwen-Long(Wan et al., 2025a). We also include human-written stories for about 250 premises as candidates to ensure real-world coverage. (2) Preference judgment, where we design a two-stage semi-automated annotation process. We use majority voting among four advanced LLMs, including GLM-4.5 (Team et al., 2025), Qwen2.5-Max (Yang et al., 2025), Claude-3.5-Sonnet (Claude, 2024), and Deepseek-R1 (DeepSeek-AI et al., 2025), to select a chosen story, followed by human verification. For cases of strong disagreement, the preferences are directly annotated by humans. In total, STORYRMB comprises $1,133$ high-quality and human-verified instances, each containing a premise, one chosen story, and three rejected stories. We then evaluate several advanced reward models on STORYRMB. We find that existing reward models struggle to reliably select human-preferred stories, with the highest accuracy at only $66.3\%$. We also find that existing reward models tend to favor LLM-generated over human-written stories, making it difficult to guide models toward human-level story generation (Tian et al., 2024).

Consequently, we propose STORYREWARD, an advanced reward model for story preferences, trained upon a large-scale, high-quality dataset of story preference pairs. To ensure real-world coverage and accurately capture human preferences, we collect a large number of human-written stories along with corresponding human preference signals. Specifically, we collect stories and associated metadata, such as category and upvote counts, from Chinese and English online literary platforms, and construct preference pairs using three methods: (1) Premise back-generation. We randomly sample story pairs from the same category with available upvote information and use an LLM to back-translate corresponding premises (Qi et al., 2024). The story with more upvotes is denoted as chosen, and the other as rejected, yielding about $6,000$ preference pairs. (2) Prompt-guided rewriting. We randomly select a premise and its corresponding human-written story as chosen, then adopt an LLM to rewrite the story as rejected. We ensure the rewriting is constrained via prompts to limit modifications, e.g., to the ending or title, preventing trivial discrepancies between chosen and rejected stories. This approach assumes human-written stories are generally preferred, which we also validate on 20 samples. This results in about $30,000$ preference pairs. (3) Human-guided continuation. We sample human-written stories and use their first part as a premise for continuation. We adopt the original human-written continuation to guide the LLM's output, which is treated as chosen, while the LLM's continuation generated directly from the premise is treated as rejected. We do not use the human-written continuation directly as chosen to avoid trivial discrepancies, as its quality is significantly higher than the LLM's continuation. This yields about $10,000$ preference pairs. We also adopt a widely-used preference-pair construction method (Cui et al., 2024). We sample two stories for a given premise and adopt an LLM to select the best as chosen. In total, we obtain $100,000$ high-quality preference pairs capturing diverse, real-world human preferences, which are then used to train STORYREWARD. STORYREWARD achieves state-of-the-art performance on STORYRMB, even surpassing much larger models. We further apply STORYREWARD in test-time scaling, conducting Best-of-N (BoN) experiments where reward models select a better story. Experimental results show that stories chosen by STORYREWARD substantially outperform those from other reward models, further validating its effectiveness in real-world applications. In conclusion, our contributions are mainly threefold:

(1) We introduce STORYRMB, the first benchmark for assessing reward models on modeling story preferences. Extensive experiments demonstrate that existing reward models struggle to capture human story preferences.

(2) We propose an automated method for collecting story preference pairs and build a large-scale, high-quality training dataset that captures diverse, real-world human story preferences.

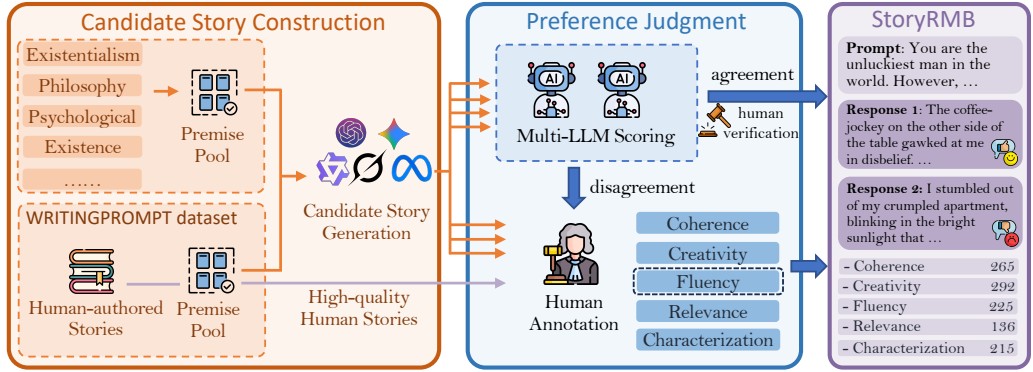

Figure 1: An overview of the benchmark construction framework. The process consists of: (1) the collection of candidate stories generated by LLMs and humans; (2) scoring the stories and partitioning them along various dimensions. These two stages yield a diverse dataset for evaluating the story reward model.

(3) We propose STORYREWARD, an advanced reward model for story preference that achieves SoTA performance on STORYRMB. We further validate its effectiveness in real-world applications.

## 2 STORYRMB: A BENCHMARK FOR STORY PREFERENCE

In this section, as illustrated in Figure 1, we present the methodology for constructing STORYRMB, including candidate story generation (§ 2.1) and a two-stage preference judgment process (§ 2.2).

### 2.1 CANDIDATE STORY CONSTRUCTION

**Prompt Collection** To generate diverse and high-quality stories, we first design a multi-stage **premise**[1] construction framework inspired by the modular synthesis approach of Ma et al. (2024). Our method begins by establishing a thematic framework drawn from literature and film, covering categories like Existentialism and Ethical Dilemmas. We then craft seed premises using a "conflict or reversal" structure (Fan et al., 2019; Barber & Kudenko, 2008) to be information-dense. These seeds are enriched with characters and settings, stylistically varied via LLM-based paraphrasing, and manually filtered for logical and cultural soundness. This process yields 624 high-quality premises. For more details, please refer to Appendix A.

In addition to the premises generated by LLMs, we add premises with human-written stories to enhance diversity. We randomly sample and filter 600 high-quality premises from the WritingPrompts dataset (Fan et al., 2018). Each of these premises is accompanied by a corresponding story authored by a human writer.

**Candidate Generation** After constructing the premise pool, we expand each premise into 4 story candidates using multiple LLMs. To ensure diversity, we select four representative models, including GPT-4o (OpenAI, 2024), Gemini 2.5 Pro (Comanici, 2025), Grok 4 (xAI, 2025), and Qwen-Long (Wan et al., 2025a). Each premise is provided to all four models under a unified prompting template, resulting in four candidate stories for each premise. These stories span a wide range of themes, languages, and narrative styles, providing a robust foundation for subsequent preference annotation. The construction details are in Appendix B.

### 2.2 PREFERENCE JUDGMENT

After collecting four story candidates for each premise, we rank them by quality, i.e., preference annotation. We first use several LLMs to score each LLM-generated story and rank the candidates accordingly. We then adopt human verification. To further identify which preference dimensions are

---

[1]*Premise* refers to the story prompt, a term widely used in the story generation literature.

most significant, e.g., whether *Story A* is preferred over *Story B* due to better coherence, we perform dimensional categorization.

**Preference Ranking**   We adopt LLMs to score each LLM-generated story and rank the candidates accordingly. We then conduct human verification for the preference rank. Following prior work (Yang & Jin, 2024), we define five scoring dimensions: creativity, coherence, fluency, characterization, and relevance. We adopt four LLMs: GLM-4.5 (Team et al., 2025), Qwen2.5-Max (Yang et al., 2025), Claude-3.5 (Claude, 2024), and DeepSeek-R1 (DeepSeek-AI et al., 2025), to provide scores for each dimension and an overall score. Based on the **overall** scores, each LLM produces a ranking of the story candidates for each premise. We then compute the agreement among the rankings from four LLMs. Specifically, given a set of $m$ candidate stories $\{c_1, c_2, \ldots, c_m\}$ and the rankings produced by $n$ LLMs, we measure the similarity between rankings using Kendall's Tau correlation coefficient (Kendall, 1938). For any two models $A$ and $B$, with ranking results denoted as $\pi_A$ and $\pi_B$, respectively, Kendall's Tau is defined as:

$$\tau(\pi_A, \pi_B) = \frac{N_c - N_d}{\binom{m}{2}}, \tau \in [-1, 1] \tag{1}$$

Let $N_c$ denote the number of concordant pairs, $N_d$ the number of discordant pairs, and $\binom{m}{2}$ the total number of possible story pairs. To assess the overall agreement among $n$ LLMs, we compute Kendall's Tau for every pair of models and take the average:

$$\tau_{\text{avg}} = \frac{2}{n(n-1)} \sum_{i<j} \tau(\pi_i, \pi_j) \tag{2}$$

If $\tau_{\text{avg}} < 0.6$, we consider it as strong disagreement and require human annotators to annotate the preference ranking. Otherwise, we use majority voting to obtain the final ranking, which is then human-verified. Additional annotation details are provided in Appendix C.

**Dimensional Categorization**   After the previous section, we have obtained the preference rankings based on the overall scores. Since the evaluation considers five dimensions, we further perform dimensional categorization to identify which preference dimensions are most significant. For example, whether *Story A* is preferred over *Story B* due to superior coherence. Following the method of prior work (Yang & Jin, 2024), we design a four-stage method to categorize each prompt by determining which dimension is most decisive in distinguishing the preferred story from the rejected candidates. The full procedure is described in Appendix F. Our final dataset contains $1,133$ preference instances. Based on the primary distinguishing dimension, the dataset is categorized into five groups: creativity (292), coherence (265), fluency (225), characterization (215), and relevance (136). The procedure of this method is as follows:

**Stage 1: Mean Gap Analysis.** For each dimension $d \in \{1, 2, \ldots, 5\}$, we calculate the mean gap:

$$\text{mean gap}_d = \text{score}_{\text{chosen}, d} - \frac{1}{|R|} \sum_{r \in R} \text{score}_{r, d} \tag{3}$$

where $R$ denotes the set of rejected stories. If one dimension exhibits a substantially larger mean gap than others (exceeding a predefined tie tolerance $\tau = 0.5$), we categorize the prompt group into that dimension.

**Stage 2: Margin to Runner-up.** When multiple dimensions show similar mean gaps, we examine the competitive margin:

$$\text{margin}_d = \text{score}_{\text{chosen}, d} - \max_{r \in R} \text{score}_{r, d} \tag{4}$$

This identifies the dimension where the chosen story has the clearest separation from the best rejected alternative. We select the dimension with the largest margin.

**Stage 3: Rejected Variance.** If margins are tied, we consider the variance of rejected scores:

$$\text{variance}_d = \text{Var}\left(\{\text{score}_{r, d} : r \in R\}\right) \tag{5}$$

We select the dimension with the smallest variance, indicating consistent poor performance among rejected stories on this dimension.

**Stage 4: Ordinal Fallback.** If ties persist across all stages, we apply deterministic ordinal ranking $(d_1 \rightarrow d_2 \rightarrow \ldots \rightarrow d_5)$ to ensure categorical assignment.

# 3 STORYREWARD: A REWARD MODEL FOR STORY PREFERENCE

This section introduces the development of STORYREWARD, including the construction of large-scale high-quality story preference pairs (§ 3.1) and the training of STORYREWARD (§ 3.2).

## 3.1 COLLECTING PREFERENCE PAIRS

To ensure real-world coverage of the collected preference pairs and reduce human annotation costs, we design an automated data collection framework to gather human-written stories and various human preferences. Specifically, we first crawl a large number of human-authored stories from popular Chinese (Douban[2]) and English (WritingPrompts[3]) online literary platforms, along with associated metadata such as categories and upvote counts. We then design three methods to automatically collect preference pairs.

**Premise Back-generation** We collect a large number of stories from the Douban website. To collect pairwise data, we first cluster stories based on similarity, where stories within the same cluster either share platform-defined genre labels or originate from the same author's column. Such clusters typically exhibit similarity in theme, writing style, and subject matter. Within each cluster, we construct story pairs by sampling two different stories and providing them as input to GPT-4o (OpenAI, 2024), which generates a premise based on the commonalities of the two stories. For preference judgment, we leverage user engagement statistics: specifically, the story with a higher readership count is considered as the "chosen". To ensure reliability, we filter out pairs with extremely low engagement, ultimately yielding approximately $6,000$ preference pairs derived entirely from authentic human-written stories. More details are provided in Appendix E.2.

**Prompt-Guided Rewriting** We select a human-written story and its corresponding premise as the "chosen" example. We then generate the "rejected" counterpart by prompting an LLM to rewrite the original story. Crucially, this rewriting process is not arbitrary. We employ a set of carefully designed prompt templates to systematically control the modifications and target specific aspects of narrative quality. These prompts constrain the LLM's changes. For instance, by instructing it to alter only the story's ending, it can undermine plot coherence. Further details are provided in Appendix E.3. This constrained approach prevents trivial discrepancies between the chosen and rejected stories. Instead, it produces rejected samples that remain readable but are deliberately flawed in their style, tone, or logic. This method is based on the assumption that human-written stories are generally of higher quality, an assumption we also validate on 20 samples.

**Human-Guided Continuation** In preliminary experiments, we observe that directly contrasting human-written stories with LLM-generated ones during reward model training often leads to rapid convergence and suboptimal performance. This is primarily due to the substantial quality gap and surface differences between human-written and LLM-generated stories. To mitigate this issue, we provide two LLMs with identical premises and prompts, and only one LLM can access the genuine human continuation of the story. After generating stories from both settings, we construct preference pairs under the hypothesis that the model guided by human-authored continuations is more likely to produce superior narratives. To validate this hypothesis, we randomly sample 20 preference pairs and determine that the model guided by human-authored continuations performs better. More generation details and prompts are provided in Appendix E.4.

In addition to constructing story preference pairs from human-written stories, we also adopt a widely used approach (Cui et al., 2024), where an LLM generates two stories for the same premise and another LLM automatically labels one as chosen and the other as rejected. More details on generation settings and prompts are provided in Appendix E.1. In total, we obtain about $100,000$ high-quality preference pairs that capture diverse and real-world human preferences, which can be directly used for reward model training.

---

[2]https://www.douban.com
[3]https://huggingface.co/datasets/euclaise/writingprompts

## 3.2 TRAINING STORYREWARD

We leverage Llama-3.1-8B-Instruct and Qwen3-8B as the base models for training. We use the premise of each instance in our collected preference pairs as the input and the stories as the preference pairs to train STORYREWARD-LLAMA and STORYREWARD-QWEN. We set the batch size to 1, the learning rate to $9 \times 10^{-6}$ for STORYREWARD-LLAMA and $2 \times 10^{-5}$ for STORYREWARD-QWEN, and train for 1 epoch.

## 4 EXPERIMENTS

### 4.1 MAIN EXPERIMENTS

**Experimental Setup** Formally, given a set of four candidate stories $\{s_1, s_2, s_3, s_4\}$, the RMs assign a score $f(s_i)$ to each candidate. A prediction is considered correct if $\arg\max_i f(s_i) = s^*$, where $s^*$ denotes the predicted chosen story. We report the proportion of correct predictions as the primary metric for evaluation. We categorize the baselines into two main groups. (1) Reward models, which are explicitly trained for reward modeling, typically formulated as regression models that assign a score to each response and select the one with the highest score. We include several advanced and representative models in this category, such as InternLM2-Reward (Cai et al., 2024), Skywork-Reward (Liu et al., 2024), ArmoRM (Wang et al., 2024a), QRM (Dorka, 2024), GRM (Yang et al., 2024), and WQRM (Chakrabarty et al., 2025). (2) LLM-based generative reward models, where large language models are employed to either directly score responses or conduct pairwise comparisons to select the preferred response (Lambert et al., 2024). In this group, we evaluate several proprietary systems, including GPT-4o (OpenAI, 2024), Gemini-2.5 Pro (Comanici, 2025), Grok (xAI, 2025), our base model Llama-3.1-8B Instruct (Meta, 2024b), and Qwen3-8B (Yang et al., 2025).

**Experimental Results** All the experimental results are presented in Table 1. We observe the following: (1) Existing reward models and LLM-as-judges perform poorly. The best model GPT-4o achieves only 66.3%, which is far from applicable in downstream applications. This indicates current reward models neglect the modeling of story preferences and further demonstrates the complexity of story preference, which requires dedicated data and modeling rather than relying on general-domain preference generalization. (2) In general, reward models perform better on the *fluency* dimension, tending to favor more fluent stories. This is intuitive, since both reward models and LLM-as-a-judge are inherently sensitive to surface fluency (Tian et al., 2024). In contrast, for more complex dimensions requiring deeper narrative understanding, such as *creativity* and *characterization*, the scores are generally lower. This underscores the inability of existing reward models to capture story-specific preference and highlights the necessity of a dedicated reward model for stories. (3) Our trained models (STORYREWARD-QWEN and STORYREWARD-LLAMA) achieve the best results on STORYRMB, significantly outperforming the base models (Llama3.1 and Qwen3), particularly on story-specific metrics such as *creativity* and *characterization*. This demonstrates the effectiveness of our training data. As our data construction method is scalable, we also encourage the community to collect more high-quality corpora to develop more advanced reward models.

### 4.2 PREFERENCE ANALYSIS: HUMAN VS. LLM STORIES

Previous studies have shown that LLM-generated stories differ largely from human-authored ones (Tian et al., 2024), remaining far from human standards. We argue that a key reason is that existing reward models often favor LLM-generated stories. To investigate this, we analyze current reward models. As noted in § 2, STORYRMB consists of two subsets: LLM–LLM pairs, where both chosen and rejected stories are generated by LLMs, and Human–LLM pairs, where the chosen story is human-written and the rejected one is LLM-generated. We evaluate the reward models on these two subsets and the results are shown in Figure 2. We find that existing models perform significantly worse on Human–LLM pairs, indicating a tendency to prefer LLM-generated stories. One possible reason is that LLMs favor LLM-generated content (Liu et al., 2023). Consequently, such reward models often fail to guide models toward producing human-like stories. In contrast, STORYREWARD performs better on Human–LLM pairs, demonstrating stronger alignment with human preferences. This advantage stems from our effective collection of human-written preference

Table 1: Accuracy (%) of invesitgated models and STORYREWARD on STORYRMB. *Average* represents the average score over the five dimensions. **Bold**: the best result. underline: the second-best result.

| Model | Coherence | Creativity | Characterization | Fluency | Relevance | Average |
|---|---|---|---|---|---|---|
| InternLM2-20B-Reward | 17.0 | 25.1 | 21.9 | 17.3 | 41.2 | 21.3 |
| InternLM2-7B-Reward | 18.5 | 18.9 | 18.1 | 15.6 | 25.0 | 18.7 |
| Skywork-Reward-Gemma-2-27B-v0.2 | 36.6 | 29.8 | 22.8 | 40.0 | 34.6 | 32.7 |
| Skywork-Reward-Llama-3.1-8B-v0.2 | 29.4 | 32.3 | 26.5 | 36.0 | 30.9 | 31.0 |
| QRM-Llama3.1-8B-v2 | 15.1 | 17.9 | 9.3 | 13.3 | 19.9 | 14.9 |
| ArmoRM-Llama3-8B-v0.1 | 52.4 | 48.8 | 44.2 | 56.4 | 43.4 | 49.7 |
| GRM-Llama3.1-8B-rewardmodel-ft | 35.1 | 28.1 | 23.7 | 35.1 | 35.3 | 31.1 |
| WQRM | 16.6 | 27.1 | 24.2 | 31.1 | 19.9 | 24.0 |
| Llama3.1-8B-Instruct | 22.3 | 30.5 | 25.1 | 16.9 | 24.9 | 24.3 |
| Qwen3-8B | 55.8 | 53.1 | 47.0 | 60.4 | 48.5 | 53.5 |
| GPT-4o | 63.4 | 63.7 | 66.0 | 73.3 | 67.6 | 66.3 |
| Gemini-2.5-Pro | 57.7 | 51.7 | 57.2 | 64.4 | 52.2 | 56.8 |
| Grok-3 | 62.6 | 57.5 | 62.8 | 69.8 | 60.3 | 62.4 |
| STORYREWARD-LLAMA | 64.5 | 60.8 | 62.3 | 53.8 | 67.7 | 61.4 |
| STORYREWARD-QWEN | **77.5** | **78.8** | **71.6** | **74.2** | **69.1** | **75.0** |

pairs. Therefore, we call on the community to leverage more human-written data in training and preference modeling to advance toward truly human-level story generation.

## 5 TEST-TIME SCALING WITH BEST-OF-N SAMPLING

An important application of reward models is test-time scaling, which allows the model to generate multiple responses and use reward models to select the best one, i.e., Best-of-N (BoN) sampling (Brown et al., 2024; Peng et al., 2025). In this section, we apply reward models in test-time scaling applications to evaluate the effectiveness of STORYREWARD. Specifically, we use the commonly used story premise dataset, the MoPS dataset (Ma et al., 2024), which contains approximately 100 premises. For each premise, we generate 16 stories using Llama-3.1-70B-Instruct (Meta, 2024a) with a sampling temperature of 1. Human annotators then rank the 16 stories according to preference, where two annotators perform rankings independently and any inconsistencies are judged by another senior annotator. We then adopt different reward models to select the best story among the 16 stories. We conduct a head-to-head evaluation, where the winning story is the one ranked higher in the human preference ordering.

Head-to-head results are shown in Figure 3. We observe that STORYREWARD substantially outperforms other reward models, with fewer than 30% of cases selecting an inferior story. This demonstrates that our models are more effective at identifying better stories during test-time scaling. Notably, our models are only of the 8B scale with low deployment and inference costs, which highlights their practicality for real-world test-time scaling. We also encourage the community to adopt our reward models for more fine-grained search, such as MCTS (Xie et al., 2024), enabling models to select even better stories.

## 6 RELATED WORK

### 6.1 STORY GENERATION

Story generation has advanced significantly with LLMs, prompting growing research into improving narrative quality. Huot et al. (2025) propose Agents' Room, a generation framework inspired by narrative theory, that decomposes narrative writing into subtasks tackled by specialized agents. Wang et al. (2024b) propose Dynamic Hierarchical Outlining with Memory-Enhancement long-form story generation method, named DOME, to generate the long-form story with coherent content and plot. Wang & Kreminski (2024) propose to use a higher-level and more abstract symbolic specification of

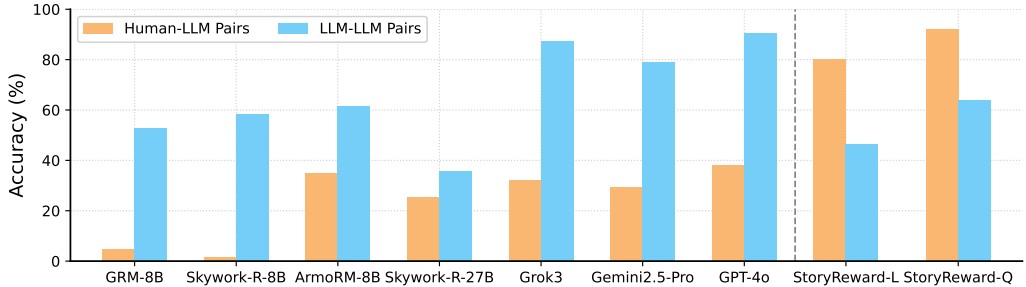

Figure 2: Accuracies (%) on LLM-LLM and Human-LLM pairs. LLM-LLM denotes all stories are generated by LLMs. Human-LLM denotes the chosen is human-written and the rejected is LLM-generated.

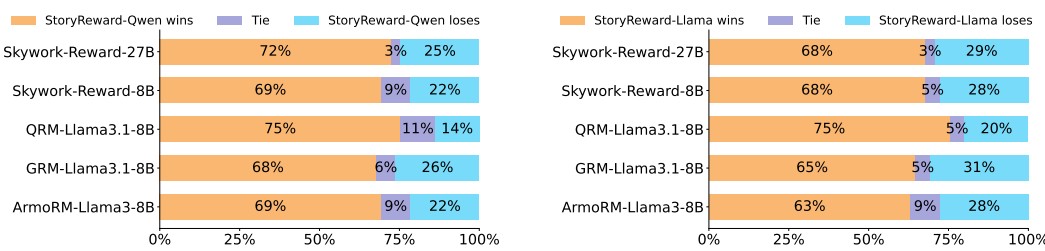

Figure 3: Head-to-head comparison results of BoN on the MoPS test set. Both STORYREWARD-QWEN and STORYREWARD-LLAMA show a dominant win rate. "Tie" means both models select the same story.

high-level story structure, implemented via answer set programming, to guide and diversify LLM-based story generation. Lee et al. (2025) propose WritingPath, a framework that uses explicit outlines to guide LLMs in generating goal-oriented, high-quality text, drawing inspiration from structured writing planning and reasoning paths, focusing on reflecting user intentions throughout the writing process. Wan et al. (2025b) introduces CogWriter to transforms LLM constrained long-form text generation into a systematic cognitive writing paradigm. CogWriter consists of a Planning Agent that performs hierarchical planning to decompose the task and multiple Generation Agents that execute these plans in parallel. However, some studies argue that LLM-generated stories remain far from human-level quality (Chakrabarty et al., 2024; Tian et al., 2024; Wang & Kreminski, 2025), which we attribute in part to the lack of effective reward model guidance during training.

## 6.2 REWARD MODELING IN STORY GENERATION

LLMs have achieved remarkable breakthroughs in tasks with objectively verifiable solutions, such as mathematics and programming. These advances have been largely driven by Reinforcement Learning with Verifiable Rewards (RLVR), which leverages reference signals derived from rule-based verifiers to provide binary feedback (correct or incorrect). Such a paradigm is highly effective when tasks admit clear ground-truth answers. By contrast, writing tasks lack standardized solutions and are inherently subjective, making them far more challenging. LLM-generated texts are often verbose and hard to verify their quality. To address this challenge, Jia et al. (2025) introduces a new training paradigm designed for non-verifiable tasks like creative writing. Instead of relying on fixed human-written datasets, Jia et al. (2025) employs a dual-agent framework: a writing agent generates stories while a reviewing agent provides detailed, constructive feedback. This feedback serves as a rich, scalable reward signal, enabling iterative self-improvement without extensive human supervision. Although not specifically targeted at story generation, this framework represents an important step toward building reward models that capture the subtleties of open-ended creative tasks. Parallel to this, Fein et al. (2025) has emerged as the first standardized benchmark and paired dataset for preference modeling in creative writing. By curating data collected from Reddit, Fein et al. (2025) provides

$43,827$ human-labeled preference pairs, establishing a foundation for training and evaluating reward models.

## 7 CONCLUSION

In this paper, we systematically evaluate and train reward models for story generation. Specifically, we propose STORYRMB, the first benchmark for evaluating reward models on story preferences. We assess several representative reward models and LLM-as-judges. We find that current reward models poorly capture human story preferences and often favor LLM-generated over human-written stories. To address this gap, we introduce STORYREWARD, an advanced reward model for story preference, trained on a large-scale, high-quality dataset collected via our automated pipeline, which gathers human-written stories and real human preference signals, and applies techniques such as rewriting for diverse coverage. STORYREWARD achieves state-of-the-art performance on STORYRMB. We also adopt reward models in test-time scaling applications, using them for BoN searching and find STORYREWARD generally chooses better stories. We suggest that the research community develop enhanced story reward models to achieve truly human-level story generation.

## ACKNOWLEDGMENTS

This work is supported by the National Natural Science Foundation of China (No. 52539001), and Beijing Natural Science Foundation (L243006). It also got partial support from National Engineering Laboratory for Cyberlearning and Intelligent Technology.

## ETHICAL CONSIDERATIONS

(1) **Intellectual property.** In our research, we utilized stories sourced from the Douban website and WritingPrompts. For the former, we strictly adhered to its copyright agreement. The latter is open-source, and we have complied fully with its licensing terms. We believe that all datasets have been properly desensitized and do not contain any personal information of the annotators or the original authors. We further adopt GPT-4o to check all human-written stories to flag potential sensitive content such as violence or explicit material. Actually, we find that there are no stories flagged as sensitive. We also randomly sample and check 10 human-written stories and find no sensitive content. For the Large Language Models (LLMs) investigated in this study, we queried LLMs (e.g., GPT-4o, Gemini 2.5 Pro, etc.) through their paid APIs. (2) **Intended Use.** This paper investigates the performance of reward models in the context of story generation and provides a high-quality benchmark and a reward model. Through the in-depth analysis presented, we aim to offer meaningful insights into LLMs and the task of story generation for the academic community. (3) **Misuse risks.** The reward model and benchmark presented in this paper are intended to improve story generation. No one should use our story generation method to create illegal stories, disseminate illicit content, or for commercial profit. (4) **AI assistance.** We used LLMs to generate data in our paper which have been stated in 2 and 3. We have used ChatGPT to refine some sentences. (5) **Worker Treatments** are discussed in Appendix C.2

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

APPENDICES

## A  PREMISE GENERATION DETAILS

We curated a diverse set of seed premises spanning multiple thematic domains. To avoid redundancy and maintain structural coherence, we introduced a thematic classification framework informed by motifs from literature, philosophy, and film. The framework organizes premises into the following categories: **Existentialism/Philosophy:** addressing fundamental questions of self, existence, meaning, and cognition. **Socio-Political Issues:** focusing on themes of power, truth, morality, and social structures. **Psychological/Consciousness-Oriented Narratives:** exploring memory, dreams, identity, and perception. **Ethical Dilemmas:** presenting morally challenging scenarios that involve difficult trade-offs. **Time and Existence:** engaging with metaphysical settings such as temporality, fate, history, and immortality. Each seed premise is designed based on two core principles:

(1) **Conflict or Reversal.** Every premise centers on a critical conflict or epistemic reversal, e.g., a philosopher discovers that their life's work rests on fabricated evidence or a historian uncovers systematic manipulation of history. Such constructs encourage LLMs to explore diverse, high-stakes narrative trajectories.

(2) **Concise Narrative Structure.** Premises follow a compact schema of subject + action/change + discovery/dilemma. Examples include a therapist experiences patients' traumas as their own or a doctor discovers a cure that requires sacrificing one life to save many. This structure provides strong narrative hooks while remaining information-dense.

To avoid overly abstract or schematic seed premises, we expanded these seeds with additional narrative elements (e.g., characters, settings, temporal contexts). For example, a philosopher discovers that their life's work is based on a fabricated premise was enriched into in a 19th-century European university town, a philosopher realizes that the foundation of their lifelong research rests on falsified historical records. This strategy preserves the thematic core while improving narrative specificity.

We further leveraged LLMs for lightweight rewriting, using structured prompts to generate stylistically varied variants. Post-processing then removed redundant or overly literal outputs, ensuring diversity and depth.

Through this procedure, we constructed an initial pool of $1,856$ candidate premises. Example entries include "a scientist proposes a theorem proving that free will is an illusion and faces backlash from multiple sides" and "a therapist, after a medical accident, begins to experience their patients' traumas as their own memories." We then performed manual filtering to remove premises that were semantically empty, logically inconsistent, thematically redundant, or culturally sensitive. The resulting collection consists of $1,000$ high-quality premises, each expressed in one to two sentences, semantically distinct, and sufficiently rich to drive story generation tasks.

## B  STORY GENERATION DETAILS

As shown in Table 3, we generate stories using LLMs as our candidate stories for bench construction. For the 533 LLM-generated premises, we adopt four LLMs to generate four candidate stories for each premise. For the 600 premises sampled from the WritingPrompts test set, we generate three candidate stories from three different LLMs, and include the original human-written story, also resulting in four candidates in total.

## C  HUMAN SCORING DETAILS

For the 600 instances containing one human-written story and three LLM-generated stories, we apply the preference ranking procedure in § 2.2 to rank only the LLM-generated stories. For the human-written stories, we filter out very short stories, i.e., less than 100 words. Given the strict quality control of WritingPrompts with our additional filtering, we consider the human-written story as the highest-quality candidate among the four candidates. We conduct further verification and ask annotators to only verify whether the human-written story is indeed the best, without needing to judge the ranking of other stories. If the annotators are uncertain, they mark the instance as "unsure".

Finally, 91 instances are labeled as "unsure" and are dropped, resulting in 509 instances, each of which contains one human-written story and three LLM-generated stories. For the annotation of 230 instances with $\tau_{\text{avg}} < 0.6$, we conduct full human annotation, detailed in Section C.1. For the 394 instances with $\tau_{\text{avg}} \geq 0.6$, we conduct human verification. We first perform major voting, where we average the overall scores from the four LLMs to produce a final score for each story candidate. The average final scores result in the final ranking. Then, we provide the annotators with the premise and the ranked stories and ask them to confirm whether the higher-ranked stories are indeed better. If the annotator can not decide, we retain the original ranking. This human-verification process reduces annotation time by roughly 30% compared to full human annotation. The choice of the 0.6 threshold for $\tau_{\text{avg}}$ is based on an empirical heuristic: $\tau_{\text{avg}} = 0.6$ means approximately 80% concordant pairs and 20% discordant pairs, which we think is sufficiently high ranking agreement.

## C.1 ANNOTATION INSTRUCTION

We implemented a local annotation platform to collect human judgments of story quality. We recruited and instructed annotators to evaluate candidate stories generated under the same premise. Annotators were instructed to read all candidate continuations for a given premise before assigning scores, and to ensure that their ratings clearly reflected differences in story quality. To maintain annotation reliability, we conducted periodic quality checks: after every 50 annotation groups, a random sample was reviewed to verify consistency and correctness.

## C.2 ANNOTATION COORDINATION

We recruited students majoring in English to serve as annotators for this project. The demographic distribution of the annotators was 58% female and 42% male. All individuals involved in the annotation process held a bachelor's degree. All staff members were compensated fairly based on an agreed-upon salary and workload. Formal contracts were signed with all annotators, and all employment practices adhered to local labor regulations. The privacy of the annotators was strictly protected, and no personal information was used for any purpose. The total cost for the annotation process, which included the annotation of stories, as well as the development and maintenance of the annotation platform, was approximately 10,000 USD.

## D HUMAN-LLM BENCH DETAILS

Table 7 shows how we generate stories using LLMs to compare with human stories.

## E TRAINING PAIRS GENERATION DETAILS

### E.1 LLMS PAIRS GENERATION DETAILS

Table 3 shows prompt we use to generate story pairs with Llama3.1-8B-Instruct (Meta, 2024b) vs Llama -70B-Instruct (Meta, 2024a), DeepSeek-R1-Distill-Llama-8B (DeepSeek-AI et al., 2025) vs DeepSeek-R1-Distill-Llama-70B (DeepSeek-AI et al., 2025), and Qwen2.5-14B (Bai et al., 2023) vs QwQ-32B (Qwen et al., 2025). On one hand we prompt larger LLMs to evaluate two stories generated by smaller LLMs, on the other hand we compare a story from a larger LLM with a story from a smaller LLM.

### E.2 PREMISE BACK-GENERATION DETAILS

In addition to story rewriting and unguided generation, we also designed prompts(as shown in Table 4) that elicit premises from existing texts. Specifically, we provide two article titles and abstracts as input and ask the model to identify their shared elements, such as themes, settings, characters, or narrative developments, and condense them into a single premise. This premise is required to be sufficiently general yet logically consistent, such that both articles can be derived from it. This procedure allows us to systematically expand the premise pool with diverse and semantically grounded entries, while maintaining narrative coherence across generated stories.

### E.3 REWRITING AS REJECTED STORY DETAILS

As shown in Table 6, to construct reliable preference pairs, we treat human-written stories as positive examples and generate low-quality negatives through controlled LLM rewriting. Specifically, we deliberately degrade text quality by altering character motivations, modifying key plot points, disrupting causal coherence, or simplifying language, thereby producing versions that remain broadly consistent with the original events but exhibit weaker narrative or stylistic qualities. These negatives are then paired with the human positives to form pairs for training the reward model. To ensure that the negatives are indeed of lower quality than the human versions, we performed random human checks on the generated outputs.

### E.4 HUMAN-GUIDED CONTINUATION DETAILS

In fact, our initial method is to use the exact human-written stories as the chosen responses. However, the trained reward model performs poorly, with an average accuracy on StoryRMB below 30%. This may be due to that the distributional gap between human and LLM-generated stories is large (Tian et al., 2024), and the reward model can easily exploit superficial linguistic shortcuts rather than focusing on story quality. Therefore, we finally adopt the human-guided continuation approach, where the LLM continues from the human-provided context. We provide two representative prompt templates as shown in Table 5: (1) human-guided generation, where the model rewrites and extends a given story beginning while preserving its core elements; and (2) unguided generation, where the model creates an original story based solely on a title.

## F DIMENSION SELECTION ALGORITHM

Algorithm 1 aims to categorize the candidate story set into a specific preferred dimension. It employs a hierarchical process to determine which dimension most highlights the chosen story. Firstly, during Stage 1, we compute the difference ($\text{mean\_gap}_d$) between the chosen story's score and the average score of all rejected stories, selecting the dimension with the largest gap as the preferred dimension. In Stage 2, if multiple dimensions yield gaps within the predefined tolerance $\tau$ (indicating a tie), we will compute the margin ($\text{margin}_d$) between the chosen story's score and the highest score among the rejected stories, selecting the dimension with the largest margin to determine the preferred dimension. In Stage 3 should the tie persist, we will compute the variance ($\text{variance}_d$) of the rejected stories' scores to assess the consistency of their performance. The dimension with the smallest variance is selected as the preferred dimension. Finally, if the analysis across the first three stages still results in a tie, the algorithm invokes the Ordinal Fallback (Stage 4), applying a pre-defined ranking to select the final preferred dimension.

## G DATA STATISTICS

The data statistics of STORYRMB and the train dataset are shown in Table 9.

## H ADDITIONAL EXPERIMENTS

### H.1 ABLATION STUDY

The results of the ablation experiment are presented in Table 2. We analyze the impact of removing key components from the method of constructing the training dataset.

(**-Premise Back-Generation**): We generate two stories from the same prompt using two models with different scales, where the story generated by the larger one is treated as the chosen story. These stories are seemed as supplementary training data. We then remove those training data generated by "Premise Back-generation" and add the supplementary data of the same quantity. We then leverage Qwen3-8B as the base models for training and evaluate the results on STORYRMB.

(**-Human-Guided Continuation**):Similarly, we remove the training data generated by "Human-Guided Continuation" and replace it with supplementary training data of the same quantity. We also leverage Qwen3-8B as the base models for training and evaluate the results on STORYRMB.

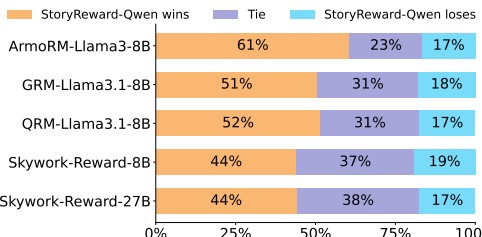 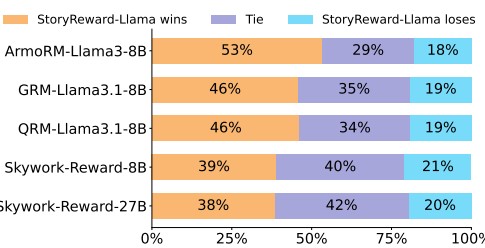

Figure 4: Head-to-head comparison results of BoN on the STORYRMB. "Tie" means both models select the same story.

**(-Prompt-Guided Rewriting)**: Be the same as the method before, we remove the training data generated by "Prompt-Guided Rewriting" and replace it with supplementary training data of the same quantity. We leverage Qwen3-8B as the base models for training and evaluate the results on STORYRMB.

## H.2 HEAD-TO-HEAD RANKING RESULTS ON STORYRMB

We further conduct an evaluation using the head-to-head ranking evaluation method in § 5 on STORYRMB. In this setting, we compare the ranking of selected stories of two reward models, where the winning story is the one ranked higher in the original preference ranking. The results are shown in Figure 4. We can observe that our model outperforms baselines, which is consistent with the findings in Figure 3. Regarding the lag LLM-LLM pairs in Figure 2, one possible reason is that while our model excels at general preference, distinguishing the absolute best candidate among high-quality, similar LLM generations remains a challenge.

## H.3 PERFORMANCE ON ANOTHER WRITING BENCHMARK

We evaluate our models on another writing benchmark. Specifically, we use the Creative Writing v3 evaluation set of EQ-Bench (Paech, 2023) with a widely-used evaluation method for story (Chhun et al., 2024). We first conduct a meta-evaluation of this evaluation method on STORYRMB, i.e., using this method to score each story and select the best one. We find it achieves $94\%$ accuracy on LLM–LLM pairs and $54\%$ on human–LLM pairs. Therefore, we believe this evaluation method is effective to evaluate LLM-generated stories.

The evaluation procedure is as follows. First, we use Llama3.1-8B-Instruct as the policy model to generate 16 candidate stories for each prompt. Then, we use different reward models to select the best story among the 16. Finally, we evaluate these selected stories using the above evaluation method. The results are reported in Table 10. We can find that stories selected by both StoryReward-Qwen and StoryReward-Llama achieve a higher score, which demonstrates the effectiveness of our trained reward models.

## H.4 LINGUISTIC ANALYSIS

We conduct linguistic analysis on stories selected by different reward models. Specifically, we adopt the concept of linguistic burstiness from quantitative linguistics to assess human-like writing. Human language is known to exhibit non-uniformity in information content and structure (Church & Gale, 1995; Katz, 1996). Following prior research (Roberts, 1996; Grieve, 2007), we use sentence-length distribution as a proxy for this phenomenon. Specifically, we quantify story burstiness via the kurtosis of sentence lengths. Its calculation formula is:

$$\text{Kurtosis} = \frac{E[(X - \mu)^4]}{\sigma^4} - 3 \tag{6}$$

$\mu$ denotes the average sentence length. $\sigma$ denotes the standard deviation of sentence length. High kurtosis indicates a frequent presence of extremely short or long sentences, which captures the structural complexity and dynamic variance typical of human writing. The results are shown in

Table 2: "(-) Premise Back-generation" removes training data contructed by "Premise Back-generation" . "(-) Prompt-Guided Rewriting" removes training data contructed by "Prompt-Guided Rewriting". "(-) Human-Guided Continuation" removes training data constructed by "Human-Guided Continuation". **Bold** indicates the best result.

| Model | Coherence | Creativity | Characterization | Fluency | Relevance | Average |
|---|---|---|---|---|---|---|
| **STORYREWARD-QWEN** | 77.5 | **78.8** | 71.6 | **74.2** | 69.1 | **75.0** |
| (-) Premise Back-Generation | **82.3** | 70.4 | **78.6** | 68.9 | **75.0** | 73.9 |
| (-) Human-Guided Continuation | 81.5 | 70.4 | 76.2 | 67.1 | 72.8 | 72.8 |
| (-) Prompt-Guided Rewriting | 73.4 | 68.1 | 71.3 | 65.2 | 68.7 | 69.9 |

Table 8. We observe that stories selected by STORYREWARD-QWEN exhibit higher kurtosis compared to the ground truth references. This suggests a potential bias: since the training data includes human-written stories, STORYREWARD-QWEN may prefer linguistic burstiness (i.e., high kurtosis) as a proxy for quality. While this provides a statistical explanation for the model's behavior, a systematic and in-depth analysis (e.g., mechanistic interpretability) requires substantial experiments beyond the scope of this paper. We leave it as future work.

---

**Algorithm 1:** Four-Stage Dimension Selection. This algorithm introduces how we detemine which preference dimension the story belongs to.

---

**Function** CategorizeDimension($M$, $\tau$, $priority$)**:**

  // Stage 1:  highest mean gaps
  $best \leftarrow score[chosen\_story]$;
  $ANS \leftarrow dim \mid max(best[dim] - M.mean[dim]) < \tau$;
  **if** $ANS$ **then**
    | **return** $ANS$
  **end**
  // Stage 2:  highest margins among Stage 1 candidates
  $second = M.max$;
  $ANS \leftarrow dim \mid max(best[dim] - second[dim]) < \tau$;
  **if** $ANS$ **then**
    | **return** $ANS$
  **end**
  // Stage 3:  lowest rejection variance among Stage 2
    candidates
  $ANS \leftarrow dim \mid |M.rej\_vars[dim]| < \tau$;
  **if** $ANS$ **then**
    | **return** $ANS$
  **end**
  // Stage 4:  use predefined priority to break ties
  **foreach** $dim$ $in$ $priority$ **do**
    **if** $dim.max == M.max$ **then**
      | **return** $dim$
    **end**
  **end**

---

Table 3: Example prompts for constructing preference pairs by contrasting stories generated by small-scale versus large-scale LLMs.

| Prompt Type | Example |
|---|---|
| How we generate premise based on two stories | Write a continuous, uninterrupted, highly detailed literary fiction story of at least {length} words, with no chapters, no scene breaks, and no meta commentary. Maintain a consistent atmospheric tone, vivid sensory descriptions, and deep psychological introspection. Premise:{prompt} The story should: – Flow continuously without numbered sections or chapter titles – Use rich imagery and internal monologue to convey the protagonist's unraveling thoughts – Gradually escalate tension without giving definitive answers to the mystery – End with an ambiguous but emotionally powerful conclusion |

Table 4: Example prompts for generating premise based on two human stories.

| Prompt Type | Example |
|---|---|
| How we generate premise based on two stories | Below are two articles. Based on their commonalities in theme, background, characters, or plot, please derive a single premise that can logically give rise to both stories. Article A:{title_a} {abstract_a} Article B:{title_b} {abstract_b} Please output a premise that can directly lead to the content of both articles. |

Table 5: Example prompts for constructing story pairs with and without human guidance.

| Prompt Type | Example |
|---|---|
| Story with Human Guidance | You are a talented fiction writer. Below is a story title and an original story beginning. Your task is to rewrite and extend the story in your own words. Preserve the original plot, characters, tone, and worldbuilding. You can improve the wording, expand on details, and make the story more vivid, but do not change the core storyline or intent. Title: {title} Original Beginning: {beginning} Rewrite the story: |
| Story without Human Guidance | You are a talented fiction writer. Below is a story title. Based on this title alone, write a complete and engaging short story. Be imaginative and creative. You can invent characters, settings, and plot freely, as long as it fits the spirit of the title. Title: title Write the story: |

Table 6: Example prompts for generating rejected story by rewriting human story.

| Prompt Type | Example |
|---|---|
| Story with Human Guidance | The following is an excerpt from a novel. Please rewrite the beginning so that it matches the ending in style and ensures consistent character motivations.
Title: {title}
Abstract: {abstract}
Content: {content}
Rewrite the beginning: |
| Story without Human Guidance | The following is a novel. Please rewrite the beginning so that it aligns with the emotional tone of the ending.
Title: {title}
Abstract: {abstract}
Content: {content}
Rewrite the beginning: |
| Story with Human Guidance | The following is an excerpt from a novel. Please rewrite the middle section so that it matches the ending in style and ensures consistent character motivations.
Title: {title}
Abstract: {abstract}
Content: {content}
Rewrite the middle section: |
| Story without Human Guidance | The following is a novel. Please rewrite the middle section so that it aligns with the emotional tone of both the beginning and the ending.
Title: {title}
Abstract: {abstract}
Content: {content}
Rewrite the middle section: |
| Story with Human Guidance | The following is an excerpt from a novel. Please rewrite the ending so that it matches the preceding style and maintains consistent character motivations.
Title: {title}
Abstract: {abstract}
Content: {content}
Rewrite the ending: |
| Story without Human Guidance | The following is the beginning and middle part of a novel. Please rewrite the ending so that it aligns with the emotional tone of the middle part.
Title: {title}
Abstract: {abstract}
Content: {content}
Rewrite the ending: |

Table 6: (continue) Example prompts for generating rejected story by rewriting human

| Prompt Type | Example |
|---|---|
| Story with Human Guidance | `The following is a passage.  Please rewrite it`
`to alter the emotional tone, while keeping the`
`events themselves unchanged.`
`Title: {title}`
`Abstract: {abstract}`
`Content: {content}`
`Rewrite the passage with a new emotional tone:` |
| Story without Human Guidance | `The following is an excerpt from a novel.  Please`
`keep the events and plot largely unchanged, but`
`modify only the final scene and outcome.`
`Title: {title}`
`Abstract: {abstract}`
`Content: {content}`
`Modify the final scene and outcome:` |
| Story with Human Guidance | `Please rewrite the following straightforward`
`narration into a literary expression that is more`
`evocative, poetic, and rhythmical.`
`Title: {title}`
`Abstract: {abstract}`
`Content: {content}`
`Rewrite in a literary style:` |
| Story without Human Guidance | `Please enhance the following passage by adding`
`the characters' psychological activities and`
`inner conflicts, making the fragment more vivid`
`and multidimensional.`
`Title: {title}`
`Abstract: {abstract}`
`Content: {content}`
`Add inner thoughts and conflicts:` |

Table 7: Example prompts for generating story with WritingPrompt Premise.

| Prompt Type | Example |
|---|---|
| How we generate premise based on two stories | Write a continuous, immersive literary fiction story of at least {length} words.  The story should unfold in a natural, human-like way--flowing as if written by a thoughtful novelist, not an AI.
Premise:  {prompt}
The story should:
Use natural pacing:  weave moments of quiet reflection with vivid external events, instead of nonstop introspection.
Balance internal monologue with dialogue and interaction, so the protagonist feels alive and connected to others.
Maintain a consistent atmospheric tone with sensory detail (sounds, smells, textures) that feel authentic rather than overly ornamental.
Prioritize human readability:  avoid mechanical repetition, vary sentence lengths, and use subtle rhythm to make the prose engaging.
Escalate tension gradually, but allow for moments of relief, small human connections, or fragile beauty.
Important:  Write as if you are crafting a novel that would be highly rated by human readers.  The prose should feel alive, compassionate, and deeply human, not robotic or overly stylized. |

Table 8: Kurtosis of story chosen by invesitgated models, STORYREWARD and STORYRMB. STORYREWARD-QWEN (on wrong predictions) represents those stories different with stories chosen by STORYRMB. STORYREWARD-QWEN (on correct predictions) represents those stories being the same as stories chosen by STORYRMB. "Difference" means relative difference of kurtosis between stories chosen by other models and stories chosen by STORYRMB.

| Model | Kurtosis | Difference (%) |
|---|---|---|
| STORYREWARD-QWEN (on wrong predictions) | 1.0149 | $+12.8$ |
| STORYREWARD-QWEN (all) | 1.0064 | $+11.8$ |
| STORYREWARD-QWEN (on correct predictions) | 0.9958 | $+10.7$ |
| GRM-Llama3.1-8B-rewardmodel-ft | 0.9573 | $+6.4$ |
| Skywork-Reward-Llama-3.1-8B-v0.2 | 0.9220 | $+2.4$ |
| QRM-Llama3.1-8B-v2 | 0.9056 | $+0.6$ |
| STORYRMB (ground truth chosen story) | 0.8999 | 0.0 |
| Skywork-Reward-Gemma-2-27B-v0.2 | 0.7598 | $-15.6$ |
| ArmoRM-Llama3-8B-v0.1 | 0.6876 | $-23.6$ |

Table 9: Dataset statistics for our training data and STORYRMB.

| Dataset | #Instances | Source | Average Length (words) | Median Length (words) |
|---|---|---|---|---|
| Training data | $93,729$ | Douban, WritingPrompts, LLMs | $2,802$ | $1,858$ |
| STORYRMB | $1,133$ | WritingPrompts, LLMs | $3,011$ | $2,401$ |

Table 10: Results on Creative Writing v3 test set of EQ-Bench (Paech, 2023) with a widely-used evaluation method for story generation (Chhun et al., 2024).

| Model | Avg Score |
| --- | --- |
| STORYREWARD-QWEN | 2.72 |
| STORYREWARD-LLAMA | 2.51 |
| GRM-Llama3.1-8B-rewardmodel-ft | 2.24 |
| Skywork-Reward-Llama-3.1-8B-v0.2 | 2.24 |
| QRM-Llama3.1-8B-v2 | 2.31 |
| Skywork-Reward-Gemma-2-27B-v0.2 | 2.28 |
| ArmoRM-Llama3-8B-v0.1 | 2.36 |

