# OpenReview forum: "StoryAlign: Evaluating and Training Reward Models for Story Generation"
_ICLR.cc/2026/Conference — ICLR 2026 Poster_

### Official Review · Reviewer_ynpP · 2025-10-28

**Soundness:** 2
**Presentation:** 3
**Contribution:** 2
**Rating:** 4
**Confidence:** 3

**Summary:**

This paper addresses the challenge that current reward models fail to capture human story preferences. The authors introduce STORYRMB, a benchmark of 1,133 human-verified preference cases comparing stories across coherence, creativity, characterization, fluency, and relevance. They show that existing reward models and LLM-as-judge systems perform poorly on this benchmark. To improve alignment, they construct STORYREWARD, a reward model trained on ~100k story preference pairs derived from human-written stories and controlled rewrites. STORYREWARD outperforms baselines and improves Best-of-N story selection, demonstrating the need for story-specific preference modeling.

**Strengths:**

1. Data creation process is rigrous.
2. This work addresses a core gap in story generation by directly modeling human narrative preference, and contributes valuable resources to the community through a high-quality human-verified benchmark and large-scale preference data, which were previously lacking.

**Weaknesses:**

1. Training details for the reward models require clarification. It is currently unclear whether STORYREWARD is simply obtained by fine-tuning Qwen and LLaMA, or whether additional architectural or training modifications are applied.
2. The paper would benefit from including straightforward fine-tuning baselines for the classification tasks, in order to contextualize the improvements introduced by STORYREWARD.
3. While STORYREWARD performs well on human–LLM preference classification, its performance notably declines on LLM–LLM preference pairs. The paper does not provide a clear explanation for this discrepancy. Further analysis is needed to understand why the model fails to generalize to LLM-generated comparisons, and what this implies about the nature of the learned preference signal.
4. Given the degraded performance on LLM–LLM comparisons, the reliability of STORYREWARD for best-of-N sampling warrants further scrutiny. If the model struggles to distinguish higher-quality continuations among LLM outputs, it may not consistently guide the selection toward genuinely better generations.

**Questions:**

See weakness.

---

> ### Author Response · Authors · 2025-11-25
>
> ### Regarding the training method (Weakness #1 & #2)
>
> Our contribution is primarily data-centric. We employ the standard RM training method on our curated dataset to develop StoryReward. Our contributions are mainly threefold: (1) constructing a high-quality story reward benchmark; (2) developing an automated pipeline to collect high-quality human preference pairs; and (3) training an advanced StoryReward model. The data-centric contribution is widely recognized and remains a critical research focus in the community [1], including automatically collecting high-quality human preference data [2, 3].
>
> Thanks for your feedback. We believe our dataset serves as a foundation for developing novel training methods. We leave the exploration of more advanced training methods, such as alternative training objectives [4], architectures [4], and formulations [5], in future work.
>
> [1] Wang, Ke, et al. "A survey on data synthesis and augmentation for large language models." arXiv preprint arXiv:2410.12896 (2024)
>
> [2] Cui, Ganqu, et al. "Ultrafeedback: Boosting language models with high-quality feedback." ICML 2024.
>
> [3] Wang, Zhilin, et al. "HelpSteer3-Preference: Open Human-Annotated Preference Data across Diverse Tasks and Languages." Neurips D&B Track 2025.
>
> [4] Wang, Haoxiang, et al. "Interpretable preferences via multi-objective reward modeling and mixture-of-experts." arXiv preprint arXiv:2406.12845 (2024).
>
> [5] Liu, Zijun, et al. "Inference-time scaling for generalist reward modeling." arXiv preprint arXiv:2504.02495 (2025).
>
> ### Regarding why the model fails to generalize to LLM-LLM comparisons (Weakness #3)
>
> Thanks for your detailed feedback. We respond to this question in our General Response.
>
> ### Regarding the BoN performance (Weakness #4)
>
> Thanks for your detailed feedback. We respond to this question in our General Response.
>
> In brief, we also adopt the Head-to-Head Best-of-N evaluation method used in Section 5 on StoryRMB. We find that our models also achieve better performance, which is consistent with the results in Section 5. We also add an additional evaluation on a widely-used benchmark (see Appendix H.3), and our models also perform better than other reward models, which indicates that our models can select better stories. We appreciate your feedback and include these discussions and experiments in Appendix H.

---

### Official Review · Reviewer_Tm6Z · 2025-10-29

**Soundness:** 3
**Presentation:** 3
**Contribution:** 4
**Rating:** 8
**Confidence:** 3

**Summary:**

The paper addresses the problem of reward modeling for story generation and makes three main contributions. First, it introduces STORYRMB, a human-curated benchmark with 1,133 instances, each containing a premise, one chosen story, and three rejected stories. This benchmark is designed to evaluate how well reward models can capture story preferences. Each instance is created by feeding a premise (from a collected pool) to both LLMs and humans to generate full stories. These story candidates are then evaluated by another set of LLMs for preference judgments, with human annotators resolving disagreements.
Second, the paper presents an automatic method to obtain large-scale story preference pairs.
Finally, it introduces STORYREWARD, a reward model trained on the automatically collected data that achieves state-of-the-art performance on the STORYRMB benchmark.

**Strengths:**

Overall, the paper is rich in content, backed by a substantial amount of work and a solid discussion. The methodology discussions are particularly strong, and the evaluations cove the contributions thoroughly and providing convincing support for the findings. That said, there are a few smaller issues that, if addressed, would definitely strengthen the paper.

**Weaknesses:**

1. There are several design decisions that could use more justification through ablation studies. For example, the *Selection with Dimensional Categorization* section (starting around line 169) or the various methods for collecting preference pairs mentioned in Section 3.1. It’s not entirely clear how much these different approaches contribute to the final dataset or the reward model’s quality.
2. Some additional discussion in the results section would also be helpful:
    - There’s a large gap between StoryReward-Llama and StoryReward-Qwen in Table 1 that isn’t really explained.
    - The paper also misses an opportunity to use the preference reasons mentioned around lines 173–175 in the evaluation.
3. A few methodological details are missing:
    - What are the sources for the initial set of stories mentioned in line 210? Are they the same sources discussed in ETHICAL CONSIDERATIONS?
    - The algorithm described in Appendix F feels vague and could use more clarification.
    - How much overlap exists between the data used to train StoryReward and the STORYRMB benchmark?

**Questions:**

Q1: I wasn’t entirely sure which part of the paper deals with the notion of “strong disagreement” mentioned around line 69. Could the authors clarify that?

**Details Of Ethics Concerns:**

The paper does not include a discussion about the actual content of the human-generated stories, whether they were collected from “online literary platforms” or written by annotators, nor the LLM-generated ones. This raises potential (though unconfirmed) concerns about socially biased or inappropriate content. From my understanding, there do not appear to be explicit guardrails in place to prevent such issues. That said, since the benchmark itself was not accessible during the review process, I cannot make any concrete claims and only note that the possibility exists.

---

> ### Author Response · Authors · 2025-11-25
>
> ### Regarding adding an ablation Study (Weakness #1)
>
> Thanks for your feedback. We add an ablation study and the detailed response is in our General Response.
>
> ### Regarding the gap between StoryReward-Llama and StoryReward-Qwen (Weakness #2.1)
>
> Thanks for the detailed feedback. We believe the gap mainly reflects model capacity: Qwen3, as a more recent generation, has substantially stronger capabilities. Table 1 also shows the large gap between Llama 3.1 8B and Qwen3 8B before RM training, where Qwen3 outperforms Llama 3.1 by nearly 30% (24.3% vs. 53.5%). Therefore, it is reasonable that StoryReward-Qwen achieves higher performance than StoryReward-Llama.
>
> A systematic investigation into why StoryReward-Qwen largely outperforms StoryReward-Llama would indeed be valuable, but such an analysis requires additional mechanistic study beyond the scope of this work. And we leave it for future work.
>
> ### Regarding the discussion on the results of different preference dimensions (Weakness #2.2)
> Thanks for your feedback. We analyze the scores of different dimensions (lines 275-278) and find that, in general, reward models perform better on the fluency dimension than creativity and characterization, where the latter requires deeper narrative understanding. This indicates that existing reward models struggle to select better stories based on story-specific features, such as creativity, instead tending to prioritize general textual attributes such as fluency. This demonstrates the necessity of developing a dedicated story reward model.
>
> ### Regarding the story source (Weakness #3.1)
> Thanks for your feedback.
> The story sources mentioned in Line 210 are the same as the discussion in the Ethical Considerations section. Specifically, the Chinese stories are from Douban and we strictly adhered to its copyright agreement. The English stories are from the open-source WritingPrompts dataset, and we also strictly adhered to its license. We have added clarifications in the paper.
>
> ### Regarding the algorithm in Appendix F (Weakness #3.2)
>
> We apologize for the ambiguity. We have revised the expression in Appendix F to improve clarity. Thanks for your feedback. Welcome to further discussion if you have any additional questions.
>
> ### Regarding the overlap between training and evaluation (Weakness #3.3)
>
> In principle, the train–test overlap should be minimal, since the training data come from the DouBan corpus and the training split of WritingPrompts, while StoryRMB is from the WritingPrompts test split.
>
> We additionally compute the 4-gram overlap between the training and test sets. We find only 0.77% of the 4-gram in StoryRMB appear in the training data, which is low and indicates that there is little contamination.
>
> ### Regarding the term “strong disagreement” (Question #1)
>
> We apologize for the ambiguity. The detailed definition of “strong disagreement” is provided in **Section 2.2 (Preference Ranking)**. Specifically, we employ 4 different LLMs to rank 4 candidate stories for a given prompt. We then calculate the ranking similarity using Equations 1 and 2. The instances with a similarity score **below 0.6** are classified as “strong disagreement” and are annotated by humans.
>
> ### Regarding the Ethics Concerns
>
> Thank you for the feedback, and we apologize for the earlier vagueness.
>
> As noted in the Ethical Considerations section, the Chinese stories are from DouBan and the English stories from WritingPrompts. All human-written stories come directly from the content provided by these platforms. We do not employ additional annotators to write new stories. We believe the stories on both DouBan and WritingPrompts have undergone their respective moderation.
>
> Following your reminder, we adopt GPT-4o to check all human-written stories to flag potential sensitive content such as violence or explicit material. Actually, we find there are no stories flagged as sensitive. We also randomly sample and check 10 human-written stories and find no sensitive content.
>
> Thank you again for the feedback. We will release all datasets in the future. We have added this clarification to the Ethical Considerations section.

---

### Official Review · Reviewer_RnUU · 2025-11-01

**Soundness:** 3
**Presentation:** 3
**Contribution:** 3
**Rating:** 6
**Confidence:** 4

**Summary:**

This paper presents a benchmark for human story preference, consisting of 1,133 human-verified data instances, and builds a reward model for story generation based on large-scale training data obtained with the proposed automated data collection method. The contributions span in three folds: a benchmark showing that existing reward models fail to capture human story preference well, an automated method to collect large-scale preference data for training, and developed a reward model with the collected data with impressive performance.

**Strengths:**

1. The task is interesting and well-justified. This paper develops a benchmark and also training data to build a reward model for story generation.

2. The performance even with a small model is promising, especially on improving Human-LLM pairs. Experiments on best-of-n sampling denotes that the developed reward model improves test-time scaling.

3. A lot of effort seems to have been spent on human evals, and collect a fairly large dataset (although how much is LLM written is slightly unclear).

**Weaknesses:**

1. While the author argue that human story preferences are inherently subjective, this is not aligned with the goal of the reward model to produce concrete scores for stories.

2. The approach WQRM from https://arxiv.org/abs/2504.07532 should be compared. WQRM is very relevant and it collects pairwise human stories and also trains a reward model.

3. To create the data, only getting humans to annotate stories where the LLM-scoring disagrees with each other - but as they mention there are biases/flaws in existing LLM writing judgements, whether LLM-judges all like/dislike the same story incorrectly is unclear.

4. The size of training dataset, size of evaluation dataset, and size of their respective sources, length of stories, etc are missing.

5. 'Premise back-translation' doesn't really mean that; the authors create another premise from two similar stories and rank them by engagement. It could be called something else. Also, they have human-human story data from this method but it's not in Figure 2 so what the reward model's accuracy is on this set seems unclear.

6. For experiments, relatedly why do other methods do better on LLM-LLM comparisons than the developed model is unclear. Some analysis would be nice.

7. For human evaluation, 16 stories seem so many to evaluate. Not sure how meaningful any ranking is.

8. It would be good to see how the developed reward model perform on another writing benchmark, or to train a model and see improvement. Basically any real downstream application.

**Questions:**

1. Dimensional categorization section is confusing. Is it all just to figure out what preference group is the most important? Maybe just do correlations?

2. Clarification question on Figure 3 - are the 'tie' sections when the models selected the same story?

**Details Of Ethics Concerns:**

None.

---

> ### Author Response · Authors · 2025-11-25
> **Response (1/2)**
>
> Thanks for your insightful and detailed feedback!
> ### Regarding the “subjective” preferences (Weakness #1)
>
> Thank you for the insightful comment. We apologize for the confusion. In this context, "preference" does not refer to personalized taste, but rather to the widely adopted pairwise data format consisting of a prompt, a chosen response, and a rejected response [1, 2].
>
> We agree that individual story preferences can indeed be subjective. However, the primary focus of our work is modeling *general* or *population-level* preferences (or principles) of story quality rather than personalized tastes. Although people differ in specific preferences, there are common principles for a good story, such as narrative coherence, logical consistency, fluency, and creativity. These shared criteria provide a stable foundation for learning and evaluation. Actually, previous evaluation of story generation also adopts these evaluation dimensions or principles [3].
>
> Current LLMs still struggle to produce high-quality stories in this general sense [4]. Therefore, in this paper, we collect human preference votes to capture these widely-agreed “preferences” or judgments and train a reward model that can select better stories and guide LLMs toward improving general story quality. We acknowledge that modeling highly-subjective personalized or user-specific preferences is an interesting and important direction, and we leave it in future work.
>
> [1] Ouyang, Long, et al. "Training language models to follow instructions with human feedback." Advances in neural information processing systems 35 (2022): 27730-27744.
>
> [2] Cui, Ganqu, et al. "Ultrafeedback: Boosting language models with high-quality feedback." (2023).
>
> [3] Yang, Dingyi, and Qin Jin. "What makes a good story and how can we measure it? a comprehensive survey of story evaluation." arXiv preprint arXiv:2408.14622 (2024).
>
> [4] Tian, Yufei, et al. "Are large language models capable of generating human-level narratives?." arXiv preprint arXiv:2407.13248 (2024).
>
> ### Regarding adding the WQRM baseline (Weakness #2)
>
> Thank you for the reminder. We have evaluated this model and reported the results in the updated Table 1.
>
> ### Regarding the potential noise of LLM voting (Weakness #3)
>
> Among the evaluation dataset StoryRMB, 394 instances (about 35%) receive strong agreement from the four LLMs. For the 35% cases, we additionally conduct human verification, as shown in Figure 1. In the verification process, annotators are provided with the model-annotated labels and required to confirm their correctness, which reduces the annotation time by roughly 30% compared to full human annotation.
>
> Thanks for your feedback. We add the corresponding clarification in Appendix C.
>
> ### Regarding the data statistics (Weakness #4)
>
> Thank you for the feedback. We have provided the detailed data statistics in Appendix G.
>
> ### Regarding the term “Premise back-translation” (Weakness #5.1)
>
> We adopt the term “Premise back-translation” to describe the backward generation process: synthesizing an instruction based on a pair of existing responses. While the term “back-translation” stems from machine translation [5], it is also adopted in other work, such as instruction back-translation [6, 7].
>
> We appreciate your feedback. We have adopted the term ”**Premise back-generation”** as an alternative and updated the paper accordingly. We remain open to further discussion if the reviewer has other suggestions.
>
> [5] Edunov, Sergey, et al. "Understanding back-translation at scale." arXiv preprint arXiv:1808.09381 (2018).
>
> [6] Li, Xian, et al. "Self-alignment with instruction backtranslation." *arXiv preprint arXiv:2308.06259* (2023).
>
> [7] Qi, Yunjia, et al. "Constraint back-translation improves complex instruction following of large language models." Proceedings of the 34th ACM International Conference on Information and Knowledge Management. 2025.
>
> ### Regarding “human-human” story evaluation (Weakness #5.2)
>
> We apologize for the confusion. The StoryRMB benchmark does not contain human–human story pairs. StoryRMB is built from the WritingPrompts test set, where each premise has only one human-written story. Therefore, we construct only human–LLM and LLM–LLM pairs. We consider this a key feature of StoryRMB, given that existing benchmarks mainly consist of LLM-generated pairs [8, 9].
>
> In the training data, we do include human–human pairs. We find that human-human pairs contribute to improving performance on StoryRMB. We have added an ablation study in Appendix H.1, which demonstrates that each data construction method provides benefits.
>
> [8] Lambert, Nathan, et al. "Rewardbench: Evaluating reward models for language modeling." *Findings of the Association for Computational Linguistics: NAACL 2025*.
>
> [9] Liu, Yantao, et al. "Rm-bench: Benchmarking reward models of language models with subtlety and style." arXiv preprint arXiv:2410.16184 (2024).

---

> ### Author Response · Authors · 2025-11-25
> **Response (2/2)**
>
> ### Regarding the performance on LLM-LLM evaluation pairs (Weakness #6)
>
> Thanks for your detailed feedback. We respond to this question in our General Response.
>
> ### Regarding the human ranking of 16 stories (Weakness #7)
>
> Thanks for your detailed feedback. First, we apologize for the typo in line 357: MoPS contains 96 premises, approximated as “100” instead of “500”.
>
> We acknowledge the potential cognitive overload of ranking 16 stories for humans. Therefore, we employ two independent annotators. Actually, we measure their top-k overlap in their ranks, i.e., the same story ratio in top-k stories. We find the inter-annotator overlap for top-4 is 55%, which far exceeds the random baseline of 25%; and the top-8 is 76%, which also significantly exceeds the random baseline of 50%. In fact, we also observe a clear quality gap between the first 8 stories and the latter 8 stories. The inconsistencies are finally judged by another senior annotator.
>
> Empirically, we find that reward models can select candidates from the top tier. We compute the average, the 25%/50%/75% percentile of the rank of selected stories. As shown in the table below, the selected stories maintain a low mean rank and 75% percentile. Since human agreement is high within the Top-8, our evaluation effectively measures model performance. We will release this data to the community for future BoN evaluations.
>
> Furthermore, as suggested, we conduct experiments on an additional writing benchmark using standard and widely-used evaluation methods (in Appendix H.3). Our model consistently demonstrates better performance, which further validates its effectiveness.
>
> | Model | Mean | 25% percentile  | 50% percentile | 75% percentile  |
> | --- | --- | --- | --- | --- |
> | STORYREWARD-QWEN | 2.185 | 1.0 | 2.0 | 2.0 |
> | STORYREWARD-LLAMA | 2.538 | 2.0 | 2.0 | 5.0 |
> | ArmoRM-Llama3-8B-v0.1 | 2.815 | 2.0 | 3.0 | 3.0 |
> | GRM-Llama3.1-8B-rewardmodel-ft | 3.092 | 2.0 | 3.0 | 4.0 |
> | Skywork-Reward-Llama-3.1-8B-v0.2 | 3.154 | 2.0 | 3.0 | 4.0 |
> | Skywork-Reward-Gemma-2-27B-v0.2 | 3.308 | 3.0 | 3.0 | 4.0 |
> | QRM-Llama3.1-8B-v2 | 4.538 | 2.0 | 3.0 | 9.0 |
>
> ### Regarding the performance on other writing benchmarks (Weakness #8)
>
> Thanks for your suggestion. We add another evaluation. Specifically, we use the Creative Writing v3 evaluation set of EQ-Bench with a widely-used evaluation method for story [1]. We first conduct a meta-evaluation of this evaluation method on StoryRMB, i.e., using this method to score each story and select the best one. We find it achieves 94% accuracy on LLM–LLM pairs and 54% on human–LLM pairs. Therefore, we believe this evaluation method is effective to evaluate LLM-generated stories.
>
> The evaluation procedure is as follows. First, we use Llama3.1-8B-Instruct as the policy model to generate 16 candidate stories for each prompt. Then, we use different reward models to select the best story among the 16. Finally, we evaluate these selected stories using the above evaluation method. The results are reported in the table below. We can find that stories selected by both StoryReward-Qwen and StoryReward-Llama achieve higher scores, which demonstrates the effectiveness of our trained reward models. We have added these experiments to Appendix H.3.
>
> | **Model** | **Score** |
> | --- | --- |
> | STORYREWARD-QWEN | 2.72 |
> | STORYREWARD-LLAMA | 2.51 |
> | GRM-Llama3.1-8B-rewardmodel-ft | 2.24 |
> | Skywork-Reward-Llama-3.1-8B-v0.2 | 2.24 |
> | QRM-Llama3.1-8B-v2 | 2.31 |
> | Skywork-Reward-Gemma-2-27B-v0.2 | 2.28 |
> | ArmoRM-Llama3-8B-v0.1 | 2.36 |
>
> [10] Chhun, Cyril, Fabian M. Suchanek, and Chloé Clavel. "Do language models enjoy their own stories? prompting large language models for automatic story evaluation." *Transactions of the Association for Computational Linguistics* 12 (2024): 1122-1142.
>
> ### Regarding the Dimensional categorization (Question #1)
>
> We apologize for the confusion. Your understanding is correct. The dimensional categorization is used to identify which preference dimensions are most significant. Our approach follows the previous methodology [11].
>
> We do not simply compute an overall correlation because Chhun et al. (2022) show that decomposing scores into orthogonal dimensions and analyzing metrics with respect to each dimension provides a more informative and discriminative evaluation. This approach reveals which aspects of story quality each signal captures, rather than capturing all preferences into one score.
>
> Thanks for your feedback. We have added some explanations in Section 2.2 (Dimensional Categorization) to make is more clear.
>
> [11] Chhun, Cyril, et al. "Of human criteria and automatic metrics: A benchmark of the evaluation of story generation." *Proceedings of the 29th International Conference on Computational Linguistics*. 2022.
>
> ### Regarding the “tie” meaning (Question #2)
>
> The reviewer’s understanding is correct. “Tie” means that both models select the same story. We add a clarification in the caption of Figure 3.

---

### Official Review · Reviewer_t4Hr · 2025-11-01

**Soundness:** 3
**Presentation:** 3
**Contribution:** 3
**Rating:** 8
**Confidence:** 3

**Summary:**

The authors of this paper highlight a critical issue in reward models for creative tasks, in that they often struggle to accurately align to human-preferences. In this paper, the authors focus on one such creative task, story-generation, and develop a first-of-its-kind benchmark to evaluate reward models for story-generation (StoryRMB). StoryRMB is comprised of >1000 datapoints, each including one gold-standard story, and three rejected stories for a given prompt. The authors then develop a reward-model for story-generation, called STORYREWARD, trained on a custom-curated dataset of 100,000 examples. The authors compare STORYREWARD to relevant baselines to highlight two key findings: (1) STORYREWARD is able to better distinguish between human-written and LLM-generated stories, and (2) STORYREWARD outperforms existing approaches in performing a Best-of-N ranking, which is an frequent task performed by reward-models.

**Strengths:**

- The paper is well motivated. The authors clearly define their research questions and effectively setup the problem with reward models for creative tasks, specifically storytelling. This was an enjoyable paper to read.
- This paper provides a first-of-its-kind benchmark in storytelling to evaluate reward models for storytelling. An important research question in the current space of LLM-research is the performance degradation of reasoning-based models on creative tasks. StoryRMB provides a helpful dataset to test the creative-writing competencies of a model.
- The authors made intelligent and targeted design choices in their data generation procedure to collect data for STORYREWARD. Through their experiments the authors validated that these choices enabled their reward model to better identify human-written stories compared to LLM-generated stories.
- I appreciated the experimental rigor in validating the assumptions made in this paper.

**Weaknesses:**

- I noticed that more than half of the examples (54,000 examples) in the dataset were generated through a procedure proposed in prior work (Cui et al. 2024). I’m curious as to the impact of this methodology compared to the novel procedures proposed by the authors. Specifically, since premise-backtranslation and human-guided continuation only comprised of 10% and 6% of the final dataset in its entirety, I wonder how critical they are in the performance of STORYREWARD. I would have been interested in an ablation study where each data collection method was individually excluded/replaced during training to evaluate its contribution to the reward model’s performance.
- The authors state that existing reward models often prefer LLM-generated stories over human-written stories. However, during the “human-guided continuation” data collection procedure, they choose to utilize the human-continuation to direct LLM-generations rather than include the human-stories themselves. Isn’t this a counter-intuitive design choice, based on the initial assertion regarding existing reward models? Can the authors justify this choice? Did the authors run any experiments where the human-written completions themselves were included in the dataset?

**Questions:**

- The authors state that R is the set of rejected stories. It is unclear to me how this set is computed. Is R the entire set except for the chosen datapoint, or is R a set of datapoints that fall under a given threshold for the combined score. If it is the latter, in equation-1, why do the authors compute a difference with respect to only the mean of the rejected samples instead of all samples.
- In addition to consistency across models, why did the authors not compute self-consistency within the generations of each individual model?

---

> ### Author Response · Authors · 2025-11-25
>
> Thanks for your valuable and detailed comments!
> ### Regarding adding an ablation study (Weakness #1)
>
> Thanks for your feedback! We add an ablation study and the detailed response is in our General Response.
>
> ### Regarding the implementation of “human-guided continuation” (Weakness #2)
>
> This is an insightful question. In fact, our initial method is to use the exact human-written stories as the chosen responses. However, the trained reward model performs poorly, with an average accuracy on StoryRMB below 30%. After discussions and analyses, we think this is reasonable: the distributional gap between human and LLM-generated stories is large [1], and the reward model can easily exploit superficial linguistic shortcuts rather than focusing on story quality. Therefore, we finally adopt the human-guided continuation approach, where the LLM continues from the human-provided context.
>
> Thank you for the feedback. We add a clarification to Appendix E.4.
>
> ### Regarding the clarification of Equation-3 (Question #1)
>
> R is the set of rejected stories, i.e., the entire set except for the chosen story. Equation 3 focuses on the chosen-rejected gap to quantify the discriminative dimensions that determine preference. Since our dataset consists of labels (chosen vs. rejected) under the same score scale (0-10), we think measuring the direct contrast between the chosen and rejected samples is more relevant than comparing against the group mean.
>
> ### Regarding the consistency method (Question #2)
>
> If we understand correctly, the reviewer is concerned about the use of multiple LLMs voting during the construction of the StoryRMB. This multi-model voting method is inspired by previous work, e.g., Ultrafeedback [2]. Compared to self-consistency from a single model, we think consistency among multiple different LLMs is harder to obtain and thus provides higher precision. The validation of this intuition is out of our scope and we leave it for future work.  However, in all cases, the final StoryRMB is verified or annotated by humans.
>
> [1] Tian, Yufei, et al. "Are large language models capable of generating human-level narratives?" arXiv preprint arXiv:2407.13248 (2024).
>
> [2] Cui, Ganqu, et al. "Ultrafeedback: Boosting language models with high-quality feedback." (2023).

---

> > ### Comment · Reviewer_t4Hr · 2025-11-27
> > **Response to author comments**
> >
> > I believe the authors have sufficiently addressed the questions/concerns raised in my review. I retain my original assessment/score that this paper is above the acceptance threshold.

---

### Author Response · Authors · 2025-11-25
**General Response (1/2)**

We appreciate all the reviewers’ efforts and their valuable and insightful comments. We respond to two common concerns here.

### Regarding adding an ablation study on data construction methods (Reviewer t4Hr, Tm6Z)
Thanks for the valuable feedback. We add ablation studies on the three data construction methods, including premise back-generation, prompt-guided rewriting, and human-guided continuation. Specifically, we remove each method in turn and replace the corresponding data using the procedure proposed in prior work [5] to keep the total data size the same. We then retrain the model on these ablated datasets. The results are shown in the table below. We can observe that removing any single method leads to a performance drop, even when its original data proportion is below 10%. This demonstrates the importance of data diversity and the value of human-written stories for training an effective story reward model. The corresponding experiments and revisions are added to Appendix H.1.
| Model | Coherence | Creativity | Characterization | Fluency | Relevance | Average |
| --- | --- | --- | --- | --- | --- | --- |
| **STORYREWARD-QWEN** | 77.5 | **78.8** | 71.6 | **74.2** | 69.1 | **75.0** |
| (-) Premise Back-Generation | **82.3** | 70.4 | **78.6** | 68.9 | **75.0** | 73.9 |
| (-) Human-Guided Continuation | 81.5 | 70.4 | 76.2 | 67.1 | 72.8 | 72.8 |
| (-) Prompt-Guided Rewriting | 73.4 | 68.1 | 71.3 | 65.2 | 68.7 | 69.9 |

### Regarding the performance on LLM-LLM evaluation pairs (Reviewer RnUU, ynpP)
We appreciate the detailed feedback. We further analyze and investigate the performance gap compared to other reward models on LLM–LLM pairs.

First, following the head-to-head ranking evaluation method in Section 5, we find that our model actually performs better on LLM-LLM pairs using this evaluation method. Specifically, each prompt in StoryRMB contains four ranked responses. The results in Table 1 and Figure 2 report Hits@1, that is, the accuracy of identifying the best chosen candidate. We further compute the head-to-head ranking results for LLM–LLM pairs. In this setting, we compare the ranking of selected stories of two reward models, where the winning story is the one ranked higher in the original preference ranking. The results are shown in the table below. We can observe that our model outperforms baselines, which is consistent with the findings in Figure 3. We add these results to Appendix H.2. Regarding the lag in Figure 2 LLM-LLM pairs, one possible reason is that while our model excels at general preference (ranking), distinguishing the absolute best candidate among high-quality, similar LLM generations remains a challenge.
| Model | Win(%) | Tie(%) | Loss(%) |
| --- | --- | --- | --- |
| STORYREWARD-QWEN vs ArmoRM-Llama3-8B | 68.14 | 25.49 | 6.37 |
| STORYREWARD-QWEN vs QRM-Llama3.1-8B | 50.43 | 30.30 | 19.27 |
| STORYREWARD-QWEN vs GRM-Llama3.1-8B | 48.94 | 30.30 | 20.76 |
| STORYREWARD-QWEN vs Skywork-Reward-8B | 41.56 | 34.43 | 24.01 |
| STORYREWARD-QWEN vs Skywork-Reward-27B | 41.65 | 36.29 | 22.06 |

| Model | Win(%) | Tie(%) | Loss(%) |
| --- | --- | --- | --- |
| STORYREWARD-LLAMA vs ArmoRM-Llama3-8B | 55.45 | 29.86 | 14.69 |
| STORYREWARD-LLAMA vs QRM-Llama3.1-8B | 46.29 | 34.54 | 19.17 |
| STORYREWARD-LLAMA vs GRM-Llama3.1-8B | 45.45 | 34.63 | 19.91 |
| STORYREWARD-LLAMA vs Skywork-Reward-8B | 38.61 | 39.83 | 21.56 |
| STORYREWARD-LLAMA vs Skywork-Reward-27B | 38.42 | 42.11 | 19.47 |

---

> ### Author Response · Authors · 2025-11-25
> **General Response (2/2)**
>
> ### (Continue) Regarding the performance on LLM-LLM evaluation pairs (Reviewer RnUU, ynpP)
> We further conduct linguistic analysis on stories selected by different reward models. Specifically, we adopt the concept of **linguistic burstiness** from quantitative linguistics to assess human-like writing. Human language is known to exhibit non-uniformity in information content and structure [1, 2]. Following prior research [3, 4], we use sentence-length distribution as a proxy for this phenomenon. Specifically, we quantify story burstiness via the **kurtosis** of sentence lengths. High kurtosis indicates a frequent presence of extremely short or long sentences, which captures the structural complexity and dynamic variance typical of human writing. The results are shown in the table below. We observe that stories selected by StoryReward-Qwen exhibit higher kurtosis compared to the ground truth references. This suggests a potential bias: since the training data includes human-written stories, StoryReward-Qwen may prefer linguistic burstiness (i.e., high kurtosis) as a proxy for quality. While this provides a statistical explanation for the model's behavior, a systematic and in-depth analysis (e.g., mechanistic interpretability) requires substantial experiments beyond the scope of this paper. We leave it as future work.
>
> | Model | Kurtosis |
> | --- | --- |
> | STORYREWARD-QWEN (on wrong predictions) | 1.0149 |
> | STORYREWARD-QWEN (all) | 1.0064 |
> | STORYREWARD-QWEN (on correct predictions) | 0.9958 |
> | GRM-Llama3.1-8B-rewardmodel-ft | 0.9573 |
> | Skywork-Reward-Llama-3.1-8B-v0.2 | 0.9220 |
> | QRM-Llama3.1-8B-v2 | 0.9056 |
> | STORYRMB ground truth chosen story | 0.8999 |
> | Skywork-Reward-Gemma-2-27B-v0.2 | 0.7598 |
> | ArmoRM-Llama3-8B-v0.1 | 0.6876 |
>
> We believe that a better data mixture could be a potential direction for improving performance on both human–LLM and LLM–LLM pairs of StoryRMB. However, exploring and verifying the data mixture method requires extensive experiments and substantial computations. We leave this exploration as future work and encourage the community to leverage our dataset to develop advanced training methods.
>
> [1] Church, Kenneth W., and William A. Gale. "Poisson mixtures." Natural Language Engineering 1.2 (1995): 163-190.
>
> [2] Slava M. Katz. 1996. Distribution of content words and phrases in text and language modelling. Nat. Lang. Eng. 2, 1 (March 1996), 15–59.
>
> [3] ALAN ROBERTS, Rhythm in Prose and the Serial Correlation of Sentence Lengths: a Joyce Cary Case Study, *Literary and Linguistic Computing*, Volume 11, Issue 1, April 1996, Pages 33–39
>
> [4] Grieve, Jack. "Quantitative authorship attribution: An evaluation of techniques." *Literary and linguistic computing* 22.3 (2007): 251-270.
>
> [5] Cui, Ganqu, et al. "Ultrafeedback: Boosting language models with high-quality feedback." (2023).

---

### Author Response · Authors · 2025-12-03
**Author Final Remarks**

We thank all the reviewers and ACs for their time and valuable feedback. We believe we have adequately responded to all concerns and would like to briefly highlight our contributions and key discussions.

### Contributions of Our Paper

- We introduce StoryRMB, the first benchmark for assessing reward models on modeling story preferences. Extensive experiments demonstrate that existing reward models struggle to capture human story preferences.
- We propose an automated method for collecting story preference pairs and build a large-scale, high-quality training dataset that captures diverse, real-world human story preferences.
- We propose StoryReward, an advanced reward model for story preference that achieves SoTA performance on StoryRMB and advanced performance on downstream BoN creative writing tasks.

### Key Discussion #1: Benchmark Construction (Reviewer t4Hr, RnUU, Tm6Z)

The main concerns of the reviewers are (1) potential noise of LLM voting; (2) the confusion of Dimensional Categorization.

(1) Regarding potential noise of LLM voting: Among the evaluation dataset StoryRMB, 394 instances (about 35%) receive strong agreement from the four LLMs. For the 35% cases, we additionally conduct human verification, as shown in Figure 1. In the verification process, annotators are provided with the model-annotated labels and required to confirm their correctness, which reduces the annotation time by roughly 30% compared to full human annotation. We add the corresponding clarification in Appendix C.

(2) Regarding Dimensional Categorization: We employ Dimensional Categorization to identify the most significant preference dimensions for fine-grained evaluation. We adopt the method from Yang & Jin (2024), which uses a four-step procedure to distinguish these dimensions. We have revised Section 2.2 and Appendix F to clarify the details of this process.

### Key Discussion #2: Training Data Construction (Reviewer t4Hr, Tm6Z)

The main concern of the reviewers is adding an ablation study to validate the effectiveness of three data collection methods. We add ablation studies on the three data construction methods. The results are in Appendix H.1. We observe that removing any single method leads to a performance drop, which demonstrates the importance of data diversity and the value of human-written stories for training an effective story reward model.

### Key Discussion #3: Experimental Results (Reviewer RnUU, ynpP)

The main concerns of the reviewers are (1) why our reward model StoryReward performs worse than some other baselines on LLM-LLM pairs; (2) the effectiveness of BoN (Best-of-N) experiments.

(1) Regarding results on LLM-LLM evaluation pairs: We first evaluate StoryReward on StoryRMB using the same settings as the BoN experiments in Section 5 (as StoryRMB contains 4 responses per prompt). The results show that StoryReward maintains a leading performance. We further analyze potential reasons by adopting the concept of linguistic burstiness. While we believe that optimizing data mixtures or training strategies could further improve performance on LLM-LLM pairs, we leave this exploration for future work.

(2) Regarding BoN results: We have provided further clarification on the data annotation for the BoN evaluation. Additionally, we add a new evaluation in Appendix H.3. These results demonstrate that our reward model effectively selects better responses.

We once again thank all reviewers and ACs for their valuable feedback, which has helped us improve our paper.

---

### Meta-Review · Area_Chair_XVXV · 2026-01-01

**Summary:**

In this paper, the authors highlight a critical issue in reward models for creative tasks: they often struggle to align with human preferences accurately. Focusing on story generation, they develop a large-scale benchmark to evaluate reward models (StoryRMB). StoryRMB comprises>1000 datapoints, each including one gold-standard story and three rejected stories for a given prompt. The authors then develop a reward model for story generation, called STORYREWARD, trained on a custom-curated dataset of 100,000 examples. The authors compare STORYREWARD to relevant baselines to highlight two key findings: (1) STORYREWARD can better distinguish between human-written and LLM-generated stories, and (2) STORYREWARD outperforms existing approaches in performing a Best-of-N ranking, which is a frequent task performed by reward models.

The authors engaged meaningfully with reviewer feedback, they clarified ambiguities and misunderstandings, and added experiments during the rebuttal concerning: 1) the distribution of preference pairs in their training benchmark 2) the quality of the LLM-LLM story pairs and 3) the best-of-N ranking evaluation.  Based on the reviewers' comments and my own reading of the paper, I believe the submission does contribute meaningfully to the lack of benchmarks for developing reward models for creative tasks like story generation. They authors should further compare their work to the following two papers which appeared in COLM, are rather related and are not referenced at all in their submission:

- https://arxiv.org/pdf/2504.07532 (suggested by one of the reviewers)
- https://arxiv.org/abs/2503.22828 (my own suggestion)

I would also urge the authors to rewrite their related work section; it currently reads like a list of prior work, but no effort is made to contextualize it or paint a broader picture of how their work fits with prior art. Please add statistical significance to your tables (it is not possible to tell whether different reward models or human judgments are meaningfully different). Please be more precise in your description of human annotators in the appendix. You say they all had an English major, even the ones who judged Chinese stories? Give details on payment, gender distribution, native language, and also the average time it took to read a pair and make a judgment. What platform did you use to administer the annotation? Finally, there is no example of a story pair in the paper. I understand your stories are long, but your paper is dense, so please include examples in the Appendix.

**Reviewer Concerns:**

I believe the rebuttal addressed all reviewer concerns.

**Reviewer Scores:**

Two of the scores were already high (i.e., 8), I don't think the 6 would have become an 8. Possibly the 4 would have become a 6. But this is all hypothetical.

---

### Decision · Program_Chairs · 2026-01-26

Accept (Poster)